Young Zambian infants with symptomatic RSV and pertussis infections are frequently prescribed inappropriate antibiotics: a retrospective analysis

Gunning Christian E. gunning.research@gmail.com 1
Rohani Pejman 1 2 3
Mwananyanda Lawrence 4 5
Kwenda Geoffrey 6
Mupila Zacharia 5
Gill Christopher J. 4
1 Odum School of Ecology, University of Georgia , Athens , GA , United States of America
2 Center for the Ecology of Infectious Diseases; Department of Infectious Diseases, University of Georgia , Athens , GA , United States of America
3 Department of Infectious Diseases, University of Georgia , Athens , GA , United States of America
4 School of Public Health, Department of Global Health, Boston University , Boston , MA , United States of America
5 Right to Care Zambia , Lusaka , Zambia
6 School of Health Sciences, Department of Biomedical Science, University of Zambia , Lusaka , Zambia
Plavec Davor
Electronic publication date: 2023 May 11
Publication date: 2023
Volume: 11
Electronic Location ID: e15175
Received 2022 Aug 9; Accepted 2023 Mar 13
Copyright: ©2023 Gunning et al.
Copyright year: 2023
Copyright holder: Gunning et al.
License: This is an open access article distributed under the terms of the Creative Commons Attribution License, which permits unrestricted use, distribution, reproduction and adaptation in any medium and for any purpose provided that it is properly attributed. For attribution, the original author(s), title, publication source (PeerJ) and either DOI or URL of the article must be cited.
License URL: https://creativecommons.org/licenses/by/4.0/

Keywords: Pertussis, RSV, Community-acquired pneumonia, Cohort study, Africa, Antibiotics, IMCI, Empiric treatment, QPCR, Pediatric health care

Funding: The National Institutes of Health/National Institute of Allergies and Infectious Diseases R01AI133080 The Bill and Melinda Gates Foundation OPP1105094 The lab testing and analyses were supported by the National Institutes of Health/National Institute of Allergies and Infectious Diseases (R01AI133080). The initial SAMIPS study and sample library was supported by the Bill and Melinda Gates Foundation (OPP1105094). The funders had no role in study design, data collection and analysis, decision to publish, or preparation of the manuscript.

==============================
Pediatric community-acquired pneumonia (CAP) remains a pressing global health concern, particularly in low-resource settings where diagnosis and treatment rely on empiric, symptoms-based guidelines such as the WHO’s Integrated Management of Childhood Illness (IMCI). This study details the delivery of IMCI-based health care to 1,320 young infants and their mothers in a low-resource urban community in Lusaka, Zambia during 2015. Our Southern Africa Mother Infant Pertussis Study (SAMIPS) prospectively monitored a cohort of mother/infant pairs across infants’ first four months of life, recording symptoms of respiratory infection and antibiotics prescriptions (predominantly penicillins), and tested nasopharyngeal (NP) samples for respiratory syncytial virus (RSV) and Bordetella pertussis. Our retrospective analysis of the SAMIPS cohort found that symptoms and antibiotics use were more common in infants (43% and 15.7%) than in mothers (16.6% and 8%), while RSV and B. pertussis were observed at similar rates in infants (2.7% and 32.5%) and mothers (2% and 35.5%), albeit frequently at very low levels. In infants, we observed strong associations between symptoms, pathogen detection, and antibiotics use. Critically, we demonstrate that non-macrolide antibiotics were commonly prescribed for pertussis infections, some of which persisted across many weeks. We speculate that improved diagnostic specificity and/or clinician education paired with timely, appropriate treatment of pertussis could substantially reduce the burden of this disease while reducing the off-target use of penicillins.

Introduction

Pediatric community-acquired pneumonia (CAP) remains a pressing global health concern, accounting for nearly 20% of deaths in young children (WHO, 2014; Liu et al., 2015). Yet appropriate diagnosis and treatment of CAP remains challenging: viral CAP is common in children (Bradley et al., 2011), while viral and bacterial CAP are difficult to distinguish (Ruuskanen et al., 2011; Principi & Esposito, 2017; Rodrigues & Groves, 2018). And while quantitative PCR (qPCR) allows for the rapid and specific identification of many respiratory pathogens (Bradley et al., 2011; Salez et al., 2015; Guthrie et al., 2019), diagnostic capacity remains scarce in low-resource settings.

To improve child healthcare outcomes globally, the World Health Organization’s Integrated Management of Childhood Illness (IMCI) was established in the mid-1990s and has since been widely adopted and periodically updated (Gove, 1997; WHO, 2005; Grant et al., 2009; Gera et al., 2016). The modern IMCI offers empiric, symptoms-based diagnosis protocols for pediatric CAP, accompanied by treatment protocols that use inexpensive and widely available antibiotics. The treatment of uncomplicated CAP (i.e., cough or difficulty breathing and fast breathing or chest indrawing) includes a 3-day course of oral amoxicillin (WHO, 2014), a preferred front-line antibiotic due to its low cost, ease of administration, low side-effect profile, and broad efficacy (Ginsburg et al., 2013; WHO, 2014; Iroh Tam, Sadoh & Obaro, 2018).

Respiratory syncytial virus (RSV) and Bordetella pertussis are two common childhood respiratory pathogens that cause widespread and sometimes severe disease, particularly in low- and middle-income countries (LMICs) (Nair et al., 2010; Liu et al., 2015). RSV, in particular, remains a leading cause of pediatric CAP worldwide, with morbidity and mortality overwhelmingly concentrated in the very young and in LMICs (Nair et al., 2010; Collins & Melero, 2011). No licensed vaccine or chemical treatment for RSV exists, though recent advances in maternal vaccination and long-lasting monoclonal antibodies show promise as prophylactic interventions (Madhi et al., 2020; Griffin et al., 2020). Widespread pertussis vaccination has dramatically reduced infant mortality worldwide (Akmatov & Mikolajczyk, 2012; Yeung et al., 2017), though vaccine formulations and schedules vary widely between countries and income levels. Of particular note, older and more reactogenic whole-cell vaccines (wP) remain the standard of care in LMICs (Domenech de Cellès et al., 2016; Forsyth et al., 2018). Unlike acellular pertussis vaccines (aP), however, wP booster doses cannot be administered to pregnant mothers during prenatal care due to reactogenicity (Forsyth et al., 2018). This lack of maternal boosting leaves newborn infants in low-resource settings with weak or absent protection from maternally-derived antibodies during their most vulnerable first months of life. Penicillins such as amoxicillin are not recommended for pertussis (Kilgore et al., 2016). However, macrolide antibiotics have been shown to limit B. pertussis transmission, despite their poor efficacy at symptom relief (Tiwari, Murphy & Moran, 2005; von König, 2005; Carbonetti, 2016; Kilgore et al., 2016) and growing concerns about resistance (Li et al., 2019a; Kamachi et al., 2020). Critically, neither RSV nor B. pertussis are directly targeted by IMCI guidelines, which are biased towards sensitivity over specificity, since the cost of inaction can be high.

In this study, we examine pediatric and maternal health care delivery in a low-resource urban setting where IMCI guidelines direct empiric clinical treatment of respiratory infections. Based on prospective longitudinal surveillance of a cohort of 1,320 young infants and their mothers, we employ an epidemiological approach to assess the interdependence between RSV and B. pertussis, symptoms of respiratory infections, and antibiotics use in mother/infant pairs. Our goal is to quantify (A) the frequency of symptomatic RSV and pertussis infections, and (B) the associated frequency of non-macrolide antibiotic use to treat these two off-target pathogens. Further, we seek to elucidate the clinical presentation and treatment of RSV and pertussis in this population, and to compare disease presentation between infants and their mothers, whose immune histories greatly differ.

Materials and Methods

Study design

This work provides a retrospective analysis of observational data collected during the Southern Africa Mother Infant Pertussis Study (SAMIPS). SAMIPS was a longitudinal cohort study in Lusaka, Zambia that sought to enroll all healthy live births in Chawama township that occurred between March and December 2015. SAMIPS followed mother/infant pairs across infants’ first four months of life during visits to the sole public health clinic (PHC) in this community. At each of seven scheduled study visits (and additional mother-initiated visits), subjects received routine, no-cost health care from PHC clinicians who followed IMCI guidelines for symptoms-based diagnosis and treatment of acute infections.

A detailed account of study methods, including sample size considerations, eligibility criteria and enrollment, nasopharyngeal (NP) sample collection and handling, and routine infant vaccinations have been previously published (Gill et al., 2016; Gunning et al., 2020; Gill et al., 2021). We provide here a brief overview of SAMIPS methods, as previously described in Gill et al. (2021). Specifically, mother/infant pairs were recruited during their first scheduled postpartum well-child visit at approximately 1 week of age. Mothers were provided with modest incentives to join and remain in the cohort. Infant eligibility included full-term, delivery without complications, and symptom-free at enrollment. Maternal eligibility included signed consent, Chawama residency (anticipated remaining in the community during study period), known HIV status prior to delivery, and, for HIV+ mothers, treatment with prophylactic antiretroviral therapy at the time of delivery. After an initial enrollment visit, mother/infant pairs were scheduled for six routine follow-up clinic visits at 2-3-week intervals through approximately 14 weeks old (maximum, 18 weeks). Additional unscheduled clinic visits were initiated by study mothers for acute medical care as well as routine well-child care. Note that we exclude unscheduled visits from consideration here.

At each clinic visit (including enrollment), NP swab samples were obtained from mother and infant. At each visit except for enrollment, detailed records of self-reported symptoms of respiratory infection were collected by clinic staff. Pre-printed barcodes and data entry forms were used to track subjects, clinic records, and NP samples. The Xcallibre digital pen system was used to automatically record dates and times, scan barcodes, and digitize hand-written text. NP samples were retrospectively tested for RSV and B. pertussis using qPCR (below), and qPCR results were not available to clinicians for use in patient care.

For antibiotics prescriptions, the medication name and dose was recorded as free-form text, and duration in days was also recorded. For symptoms, mothers were asked if the following symptoms were present since their previous clinic visit in themselves and their infants: cough, coryza, uncontrollable coughing, whooping, posttussive vomiting, cyanosis, labored breathing, and wheezing, and (only in infants) hot/feverish, difficulty feeding, lethargy, fits or seizures, and apnea. Clinicians directly observed and documented the presence of cyanosis, paroxysmal cough, whoop, apnea, conjuctival injection, mechanical sequellae of cough, lethargy, and bronchitis or pneumonia, and (only in infants), poor suck, seizure, and chest wall in-drawing. Clinicians also measured and recorded temperature. We define fever as a temperature above 38 degrees C. Note that, while symptoms were not systematically recorded at the enrollment visit, enrollment criteria specified that infants (but not mothers) were free of apparent symptoms.

All study visits, including enrollment, occurred at the Chawama Primary Health Clinic (PHC). We note that the PHC is the only government-supported clinic in this community, as well as the only source of no-cost health care. Consequently, we expect that unrecorded, off-site healthcare utilitation during our study was rare among subjects. PHC staff included Nurse Midwives and Clinical Officers (Kamfwa, 2021), who evaluated subjects and prescribed antibiotics according to IMCI guidelines. Prescriptions were filled on-site at the time of visit.

Routine childhood vaccinations were administered by clinic staff at appropriate clinic visits according to the official Zambian schedule (for details, see Gunning et al., 2020). All vaccinations were administered after NP samples were collected and in a separate area of the clinic in order to avoid possible cross-contamination between NP samples and vaccines.

Laboratory methods

NP samples were processed as previously described in Gill et al. (2021). Specifically, NP sample DNA and RNA were extracted using the NucliSENS EasyMag system (bioMérieux, Marcy l’Etoile, France). We used quantitative PCR to test samples for B. pertussis, RSV, and human RNAse P (RNP). RNP is a constitutively expressed gene that we used to assess successful sample collection, storage, DNA extraction, and lack of PCR inhibition. The B. pertussis assay was a singleplex TaqMan qPCR reaction targeting the IS481 insertion sequence (Tatti et al., 2008), and the RSV assay used a reverse-transcriptase qPCR reaction (Wang et al., 2019). Samples were run on 96-well qPCR plates; each plate contained approximately 46 samples (one each of IS481 and RNP), along with one positive and one negative control per plate. Each plate was run for 45 cycles, such that the minimum detectable target quantity had a CT value of 45 (lower values indicates more target). We consider assays with a CT value of 45 or less to be detecting assays; all others were considered non-detecting (N.D.).

All primers and probes were purchased from Life Sciences Solutions (a subsidiary of Thermo Fisher Scientific Inc.). Most plates were run on an ABI 7500 thermocycler (Thermo Fisher Scientific Inc., Waltham, MA, USA). Starting in 2019, some plates were run on a QuantStudio5 thermocycler (Thermo Fisher Scientific Inc., Waltham, MA, USA). A subset analysis of samples run in parallel on both machines showed minimal systematic variation between machines, and we do not distinguish between machines here.

Statistical analysis

We focus here on the 1,320 mother/infant pairs who have ≥4 NP samples per subject, which is the same analysis set as in Gill et al. (2021). We have excluded unscheduled clinic visits and limit our analysis here to enrollment and scheduled clinic visits in order to avoid the potentially confounding effects of self-initiated health care and more clearly focus on routine surveillance.

We categorize any detectable qPCR signal (CT ≤45) as a detection. Recognizing that some unknown portion of these detections constitute false positives, we nonetheless have demonstrated that weak qPCR signals (IS481 CT >40) contain important epidemiological information in this cohort (Gill et al., 2021). Our goal here is population-level surveillance and epidemiological inference rather than disease diagnosis in patients. As such, our approach favors sensitivity over specificity in interrogating cohort-level associations between between pathogens and symptoms.

In order to categorize prescriptions by antibiotic mode of action, we manually inspected free-form text, and then developed a set of regular expressions that accounted for common abbreviations and optical character recognition errors. Prescription could include more than one medication and corresponding category.

Our temporal analysis used a set of generalized additive models (GAMs) to estimate the relative frequency of pathogen detection throughout the study (one model per pathogen). Our GAMs use a binomial link function, and were smoothed by calendar date using cubic regression splines with shrinkage. For inference, we used a set of logistic regression models to estimate the per-visit probability of (A) respiratory symptoms and (B) antibiotics prescription, with separate models for mothers and infants. The models that estimate antibiotics prescriptions used observations from both the current and previous visit as covariates. Initial enrollment visits were thus excluded from the antibiotics models, as antibiotics were not recorded at enrollment. We also use these logistic regression models to assess the presumed causal relationship between pathogens (RSV and pertussis), symptoms, and antibiotics use. Our analysis (and presentation order) assumes that pathogens cause symptoms, and that antibiotics were prescribed in response to these symptoms.

All analysis was conducted in R version 3.5.2 (R Core Team, 2018). We use the Mann–Whitney test (R function wilcox.test) to compare IS481 CT values between subjects with and without symptoms. We employ the Wald method (unconditional maximum likelihood estimation) in the epitools package to compute relative risk and corresponding p-values.

Results

Overview

Our analysis set includes 8,390 visits and 16,784 NP samples across 1,320 mother/infant pairs from March 2015 through Jan 2016. A high level summary of outcomes is shown in Table 1, stratified by unique subjects and visits (columns) where each outcome was observed: antibiotics prescription during visit (ABX), self-reported symptoms of respiratory infection, and detection of pertussis and/or RSV. In Fig. 1, we illustrate the co-occurrence frequency (within study visits) of each pair of outcomes, stratified by mothers versus infants (Fig. 1A), as well as concurrent outcomes between infants and their mothers (Fig. 1B).

Table 1 Frequency (and percent) of unique subjects and visits where each study outcome was observed, out of 1,320 mother/infant pairs and 8,390 regularly scheduled clinic visits (including enrollment).

In addition, six infants in the analysis set died during the study.

Outcome	Subjects	Visits	
Infants	
ABX	207 (15.7)	258 (3.1)	
Symptoms	568 (43.0)	768 (9.2)	
Pertussis	429 (32.5)	670 (8.0)	
RSV	36 (2.7)	41 (0.5)	
Pertussis & RSV	6 (0.5)	6 (0.1)	
Mothers	
ABX	106 (8.0)	124 (1.5)	
Symptoms	219 (16.6)	240 (2.9)	
Pertussis	471 (35.7)	759 (9.0)	
RSV	27 (2.0)	28 (0.3)	
Pertussis & RSV	4 (0.3)	4 (0.0)	
Notes.

ABX antibiotics prescribed during visit

Symptoms symptoms of respiratory infection

Figure 1 Co-occurrence of outcomes (rows, columns) during study visits.

(A) Percent of visits by infants (left, N = 8, 390) or mothers (right, N = 7, 074) with each outcome (columns, X) in which other outcomes (rows, Y) was also observed (Y/X). (B) Percent of infant visits with each outcome (columns) where the same outcome was also observed in the infant’s mother (Mother/Infant). Text shows percent co-occurrence and the number of visits in the numerator and denominator. NP samples were collected at all visits, but symptoms of illness (Symp) and antibiotics prescriptions (ABX) were not recorded at enrollment visits. Since only apparently healthy infants were enrolled, enrollment visits for infants but not mothers are included here.

As expected based on IMCI guidelines, both infants and mothers were commonly prescribed antibiotics during visits where respiratory symptoms were recorded (infants = 20.2% of symptomatic visits, mothers = 27.5%, Fig. 1A). Conversely, respiratory symptoms were very commonly reported during visits where antibiotics were prescribed (infants = 60.1% of visits with antibiotics prescriptions, mothers = 53.2%), suggesting that respiratory illness accounted for a large fraction of antibiotics use in this cohort.

As we previously reported (Gill et al., 2021), we detected pertussis with surprising frequency in this cohort (8.0% and 9.0% of visits for infants and mothers, Table 1). On the other hand, we detected RSV in very few subjects overall (36 infants and 27 mothers). Unsurprisingly, we observe symptoms much more frequently in infants than in mother during visits when either pertussis (21.5% vs 4.1%) or RSV (29.3% vs 5.3%) was detected. Within mother/infant pairs (Fig. 1B), we find frequent concurrent antibiotics use (20.2% of infant visits with antibiotics prescriptions), symptoms (18.8%), and pertussis detections (29.1%).

Prevalence of RSV and pertussis

In Fig. 2, we show the estimated proportion of NP samples where RSV (Fig. 2A) or pertussis (Fig. 2B) was detected across the study period. We also show NP sampling intensity over time (Fig. 2C), which reflects the cohort’s rolling enrollment. In March 2015, at the beginning of our study, we observe a brief period of elevated RSV prevalence that suggested the end of a seasonal outbreak, consistent with previous observations in Lusaka and similar locales (Nakazwe et al., 2019; Li et al., 2019b). This was followed by an outbreak of pertussis that stretched from May to August 2015, during the coolest months of the year in Lusaka. Multiple pertussis detections were common: we observed ≥2 detections in 12% of infants and 13.9% of mothers, and ≥3 detections in 4.5% of infants and 5.7% of mothers. Repeat detections of RSV, on the other hand, were rare (3 infants and 1 mother). Both RSV and pertussis were simultaneously detected in just 6 infants and 4 mothers (0.5% and 0.3%, respectively).

Figure 2 (A–B) Estimated percent NP samples (lines) over time with detected RSV (A) or IS481 (B) for mothers versus infants (color) (one GAM per line; shading shows 95% CI). Points show percent sample detections within each group (shape), binned by week. (C) NP samples per calendar week (N = 16,784).

Symptoms of respiratory infections

Symptoms were reported and/or observed in 568 infants (43%) across 768 visits (9.2%), and in 219 mothers (16.6%) across 240 visits (2.9%) (Table 1). Most symptoms were mild: simple cough and/or coryza alone accounted for 84.9% (infants) and 90.8% (mothers) of visits where symptoms were recorded. In addition, simple cough and/or coryza were sensitive indicators of other symptoms, such that only 3.5% (infants) and 5% (mothers) of visits where any symptoms were recorded included neither cough nor coryza.

As expected, visits with symptomatic pertussis detections were much more common in infants (21.5%) than in mothers (4.1%) (Fig. 1). As we previously described (Gill et al., 2016), severe pertussis was rare in this cohort and, while RSV detections were also rare, symptomatic RSV detections were much more frequent in infants than mothers (12/41 visits and 1/19 visits, respectively).

Logistic regression showed that an infants’ odds of symptoms were approximately 3-fold higher when pertussis was detected, and 4.7-fold higher when RSV was detected (Table 2). We also observed a modest increase in the odds of symptoms as infants age (OR = 1.06 per week of age). We presume this increase corresponds with infants acquiring infectious diseases across the course of the study, consistent with our previous findings of age-dependent pertussis prevalence in study infants (Gill et al., 2021). Also consistent with our previous findings, we do not observe a consistent relationship between RSV or pertussis and symptoms in mothers.

Table 2 Symptoms of respiratory infection: estimated effects of covariates on the presence of symptoms.

Separate models were constructed for infants (N = 8,390 visits) and mothers (N = 7,074 visits).

Covariate	OR	P	95% CI	
Infants	
Infant Age (wks)	1.06	<0.001	1.04–1.08	
Pertussis	2.95	<0.001	2.40–3.61	
RSV	4.69	<0.001	2.23–9.25	
Mothers	
Infant Age (wks)	0.96	0.022	0.93–0.99	
Pertussis	1.25	0.3   	0.81–1.86	
RSV	1.53	0.7   	0.08–7.45	

Most pertussis detections in this cohort exhibited weak qPCR signals that would be considered ‘negative’ in clinical settings (CT >40) (Gill et al., 2021). We did, however, observe modestly lower IS481 CT values when symptoms were present versus absent in infants (median 41.8 vs 43.4, Mann–Whitney test p < 0.001) but not in mothers (p = 0.27) (Fig. 3).

Figure 3 Distribution of IS481 CT values, stratified by the presence of symptoms (color).

In infants, CT values were 1.35 less (i.e., stronger qPCR signals) when symptoms were present (Mann-Whitney test, p < 0.001, 95% CI of difference of location: 1.0–1.7). For clarity, NP samples with CT <35 are not shown (20 out of 16,784 total NP samples).

Antibiotics use

Overall, we recorded 382 antibiotics prescriptions for 313 unique subjects. Approximately twice as many infants received an antibiotics prescription during the study as mothers (207 infants versus 106 mothers). Infants also received more antibiotics than mothers (Table 3): while a single mother received three prescriptions, ten infants received ≥3 prescriptions (max four).

Table 3 Number of unique subjects that received antibiotics, stratified by the number of clinic visits at which antibiotics were prescribed (rows).

Visits	Infants	Mothers	
1	167	89	
2	30	16	
3	9	1	
4	1	0	
Sum	207	106	

The recorded duration of most prescriptions was five days (79.8%), followed by seven days (10.2%), and not recorded (6%). Most prescriptions were for penicillins (62% and 67% in infants and mothers), followed by cephalosporins (17.1% and 3.2%), undetermined (11.6% and 21%), along with infrequent prescriptions of aminoglycosides or nitroimidazoles (2.3% and 6.5% in infants and mothers). Use of macrolides and sulfonamides were rare, accounting for 6.2% and 2.4% (macrolides) and 3.1% and 3.2% (sulfonamides) of prescriptions in infants and mothers, respectively.

In Table 4, we detail how antibiotics use in infants and mothers changed between clinic visits, and in response to pathogens. We observed a large positive association with antibiotics prescription at the previous clinic visit (infant OR = 5.4, mother OR = 6.4, p < 0.001), and a high correspondence of antibiotics prescriptions within mother/infant pairs (infant OR = 19.7 and mother OR = 25.2, p < 0.001). Albeit with very small sample sizes and weak statistical evidence, we observe that RSV detections predicted antibiotics use in infants (OR = 3.2, p = 0.055). Interestingly, we observe that contemporaneous pertussis detection corresponded with increased antibiotics use in infants (OR = 2.2, p < 0.001) but not mothers (OR = 0.6, p = 0.2). Conversely, detection of pertussis at the previous visit predicted antibiotics use in mothers (OR = 2.3, p = 0.004), with a less pronounced association in infants (OR = 1.5, p = 0.047).

Table 4 Prescription of antibiotics: estimated effect of covariates at prior or current visit, relative to prescription.

Separate models were constructed for infants (N = 7,066 visits) and mothers (N = 5,748 visits). Co-pair indicates the status of an infant’s mother and a mother’s infant, respectively.

Covariate	OR	P	95% CI	
Infants	
Prior ABX	5.4	<0.001	3.5–8.1	
Prior Pertussis	1.5	0.047	1.0–2.2	
Current Pertussis	2.2	<0.001	1.5–3.1	
Current RSV	3.2	0.055	0.8–9.4	
Current co-pair ABX	19.7	<0.001	13.1–29.4	
Mothers	
Prior ABX	6.4	<0.001	2.9–13.1	
Prior Pertussis	2.3	0.004	1.3–3.9	
Current Pertussis	0.6	0.2   	0.3–1.2	
Current RSV	9.7	0.031	0.5–51.4	
Current co-pair ABX	25.2	<0.001	16.4–38.7	
Notes.

OR odds ratio

For illustrative purposes, we display in Fig. 4 the study timeline of all mother/infant pairs where a subject received three or more antibiotics prescriptions (eleven subjects across ten pairs). Here we see that repeated antibiotics prescriptions typically form a contiguous sequence across multiple visits, and that concurrent antibiotics use in mothers and infants is common. We also highlight the infrequent prescriptions for macrolides and sulfonamides, both of which are recommended treatments for B. pertussis infections. Our results indicate persistent pertussis infections in four infants (C, E, F, and H) and three mothers (D, E, and G), and suggest that pertussis infections persist during and after antibiotics use.

Figure 4 Study timeline of all mother/infant pairs where a subject was prescribed antibiotics at ≥3 visits (pairs are ordered by enrollment date, * indicates HIV+ mothers).

Shape indicates presence of symptoms, color shows qPCR results, and numbers show cumulative antibiotics prescriptions. Arrows indicate prescription of macrolide (infants B and C) or sulfonamide (mother D) antibiotics. Clinical records were not collected at enrollment, but only infants without apparent disease were enrolled. Of ten infants with ≥3 antibiotics prescriptions, four had multiple pertussis detections (RR = 4.9, p = 0.024; infants C, E, F, and H) and two had an RSV detection (RR = 8.9, p = 0.028; infants E and F).

Discussion

In this study, we leverage intensive monitoring of mother/infant pairs during routine well-child health care across newborn infants’ first four months of life to provide a fine-grained view of realized health outcomes. Through longitudinal surveillance of mother/infant pairs, we document the intensity of antibiotic use over time in the context of routine qPCR testing for RSV and pertussis. Overall, we found that antibiotics use was widespread, particularly in infants. As expected, we observed a strong association between symptoms and antibiotics use, which commonly coincided with B. pertussis detection in infants. Antibiotics use consisted mostly of penicillins and cephalosporins, while macrolides and sulfonamides, which can effectively treat B. pertussis, were only rarely prescribed. We were surprised at the widespread detection of B. pertussis in both infants and mothers (8% and 9% of NP samples, resp.), albeit often with very weak qPCR signals (i.e., high CT values). Nonetheless, the observed correspondence between symptoms and stronger qPCR signals in infants highlights the clinical relevance of our findings.

The persistent scarcity of high-quality B. pertussis surveillance has long been recognized by clinical and public health communities, particularly across Africa (Guiso, Liese & Plotkin, 2011; Muloiwa et al., 2018). Passive B. pertussis surveillance is complicated by the low specificity of symptoms for pertussis in adults, complicating empiric treatment guidelines such as IMCI. For example, Miyashita et al. (2013) found that paroxismal cough was a highly sensitive (90%) but non-specific symptom of pertussis, while the most specific symptoms of posttussive vomiting and whoop showed low sensitivity (≤25%). Our study echoes previous findings of under-reported infant and adult pertussis in Africa (Scott et al., 2015; Moosa et al., 2019). We add to this record by carefully documenting the clinical presentation of putative infections across repeated visits, along with IMCI-based empiric treatment. Our study’s prospective design also allows us to draw conclusions about the study population at large, in contrast to more common opportunistic sampling of symptomatic children and their mothers.

A noteworthy limitation of this study is the infrequent detection of RSV, which offers limited statistical power for inference. On one hand, we expected to detect RSV more frequently than B. pertussis in infants, as no RSV vaccine is available and almost all children contract RSV by three years of age (Cox et al., 1998; Faneye et al., 2014; Nyiro et al., 2017). On the other hand, maternal antibodies provide some protection during infants’ first 1–2 months of life (Nyiro et al., 2017), while RSV’s shorter duration of infection would yield fewer detections relative to pertussis at a comparable incidence. Our results also suggest that we captured the tail of an RSV outbreak in this population while the cohort size was small but increasing (due to rolling enrollment), and while enrolled infants were no more than 2 months of age. Nonetheless, we detected RSV in approximately 2% of study mothers during this short window, highlighting the potential importance of caregivers in RSV transmission (Walsh, 2017).

The observational nature of this study is both a strength and a limitation. We cannot confidently ascribe causal relationships to the observed patterns, nor can we rule out the impact of unobserved, epidemiologically important covariates on our findings. However, by prospectively recording the delivery of pediatric care in a representative clinical setting, we were able to characterize drug-pathogen interactions that, due to logistical constraints and ethical considerations, would be difficult or impossible to evaluate in clinical trials. Similarly, our retrospective qPCR testing of NP samples precludes any clinical impact of test results, while our focus on population-level surveillance allows us to include more sensitive (and less specific) qPCR signals than would be appropriate for clinical disease diagnosis. Consequently, this study provides an incomplete but nuanced view of realized health care outcomes in a representative low-resource urban community. Indeed, we posit that our study population is broadly representative of risks and challenges facing at-risk urban communities worldwide, including rapid growth, limited access to health care, economic constraints on treatment options, and major gaps in public health disease surveillance (Satterthwaite, 2003; WHO, 2018; Connolly, Keil & Ali, 2021).

Public health impacts for low-resource settings

Trade-offs between over- and under-treatment pose a recurring dilemma in routine clinical care. This dilemma is particularly acute in low-resource settings, where staff, facilities, and diagnostic capacity are limited, and where treatment costs are constrained. A priori, we expect that public health clinicians seek to minimize ineffective use of antibiotics in order to reduce avoidable disease, side-effects, and financial costs. Yet the immediate costs of inappropriately treating, e.g., B. pertussis with amoxicillin, may be low relative to the risks of failing to treat sensitive pathogens, especially in pediatric care.

We have demonstrated the routine use of penicillins in our cohort as a non-specific (and potentially prophylactic) treatment of respiratory symptoms, particularly in infants. By pairing clinical records with molecular surveillance, our study highlights a notable gap in current IMCI guidelines regarding pertussis. Our results indicate that minimally symptomatic pertussis was frequently treated by PHC clinicians with inappropriate non-macrolide antibiotics, and that these pertussis infections commonly persisted in the absence of appropriate chemical treatment. We note that a recent IMCI update recommends the addition of an inexpensive macrolide for symptomatic children over three years of age who have completed an initial round of amoxicillin and are “not better but not worsening at time of re-assessment”. However, these updates do not directly address B. pertussis (Grant et al., 2009). We also note the virtual absence of reported pertussis cases in Lusaka, despite evidence for widespread asymptomatic and minimally symptomatic pertussis in this cohort (Gill et al., 2021). We speculate that improved diagnostic specificity and/or clinician education paired with timely, appropriate treatment of pertussis in infants and their mothers could substantially reduce the burden of disease in low-resource settings such as this while also reducing the off-target use of penicillins.

We wish to thank Rachel Pieciak, William MacLeod, Magdalene Mwale, Arash Saeidpour, Deven Gokhale, and Tobias Brett for their helpful comments. We also wish to thank the Lusaka lab team who generated the results for this analysis: Caitriona Murphy; Ruth Nkazwe; Chilufya Chikoti; and Baron Yankonde.

Additional Information and Declarations

Competing Interests

Author Contributions

Ethics

Data Availability

Zacharia Mupila and Lawrence Mwananyanda are employed by Right to Care Zambia.

Christian E. Gunning conceived and designed the experiments, performed the experiments, analyzed the data, prepared figures and/or tables, authored or reviewed drafts of the article, and approved the final draft.

Pejman Rohani conceived and designed the experiments, authored or reviewed drafts of the article, and approved the final draft.

Lawrence Mwananyanda conceived and designed the experiments, authored or reviewed drafts of the article, and approved the final draft.

Geoffrey Kwenda performed the experiments, authored or reviewed drafts of the article, and approved the final draft.

Zacharia Mupila performed the experiments, authored or reviewed drafts of the article, and approved the final draft.

Christopher J. Gill conceived and designed the experiments, performed the experiments, authored or reviewed drafts of the article, and approved the final draft.

The following information was supplied relating to ethical approvals (i.e., approving body and any reference numbers):

The Southern African Mother Infant Pertussis study (SAMIPS) was approved by the ERES Converge IRB in Lusaka, Zambia, and the Boston University Medical Center IRB (ERES Converge: 2015-Jan-002; Boston University Medical Center: H-34096). The current analyses uses fully de-identified SAMIPS data that contains no personally identifiable information.

The following information was supplied regarding data availability:

The data and code are available at OSF: Gunning, C. E. 2022. “Manuscript: Young Zambian Infants with Symptomatic RSV and Pertussis Infections Are Frequently Prescribed Inappropriate Antibiotics.” OSF. December 15. doi: 10.17605/OSF.IO/TNKYU.

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
