# Peer review of "Young Zambian infants with symptomatic RSV and pertussis infections are frequently prescribed inappropriate antibiotics: a retrospective analysis"

_PeerJ, doi:10.7717/peerj.15175_

## Round 0.1 · original submission · Major Revisions

Dear Authors,

Please make corrections as suggested by the reviewers or write a detailed rebuttal on a point-by-point basis.

Reviewer 1 ·

Basic reporting

There seems to be an insufficient amount of tables, figures and graphical data supporting the research

Experimental design

This is not an original research, but indeed a widespread issue that needs to be adressed

Validity of the findings

no comment

Additional comments

The use of inappropriate antibiotics for treating various bacterial, even viral diseases are a widespread medical issue that needs to be appropriately addressed. This article provides an insight into the actual problem which is lack of knowledge in prescribing appropriate antibiotics and staying up to date with new diagnostic and therapeutic issues.

Reviewer 2 ·

Basic reporting

No comment.

Experimental design

No comment.

Validity of the findings

No comment.

Reviewer 3 ·

Basic reporting

The manuscript has redundant paragraphs. For example, methods were reproduce in the first section of the results.

Example of typos include "logistic generalized linear models (GLMs)" on line 274.

Manuscript can be shortened by much. very long unnecessary informations.

Experimental design

- As admitted by the authors, the study design did not capture the RSV season for the region. Therefore, I was not clear on what useful information could be obtained from this study with regards to authors achieving their goal: " A) the frequency of symptomatic RSV and pertussis infections, and B) the associated frequency of non-macrolide antibiotic use to treat these two off-target pathogens."

In US, we observe about 50-60% of RSV infection in infants during peak season. How does this study inform the incidences of both RSV and B. pertussis in Southern Africa?

Validity of the findings

The study is valid. However, the contribution of this paper should be framed within use of antibiotics for viral infection and antibiotic resistance, The attempt to quantify incidence of RSV infection when the study design clearly does not capture the RSV season was not convincing.

I also don't understand why adult symptom was compared to infants. Why anyone expect adults to have worse symptom(if any)?

Additional comments

It would be helpful to conduct analysis on what leading factors were associated with antibiotics prescription for viral infections? What are the incidence of opportunistic bacterial infection in those infants with RSV and B. pertussis infections?

---

## Round 0.2 · Minor Revisions

Dear Authors,
please make corrections as suggested by reviewer#3.

Reviewer 2 ·

Basic reporting

No comment.

Experimental design

No comment.

Validity of the findings

No comment.

Reviewer 3 ·

Basic reporting

Manuscript has improvement, but not concise enough. I find authors response to be combative rather than using reviewers feedback to improve the manuscript. It is not reviewers job to proofread and edit the manuscript,

Experimental design

It is puzzling that the authors claim their own prospective cohort as retrospective analysis, The cohort enroll infants and collected data with primary aim of quantifying incidence of RSV and pertussis infection. So, what makes this particular manuscript's data analysis retrospective?

The study has cohort enrollment timing flaw for RSV season. Figure 2a is an evidence for missing RSV peak and capturing pertussis peak. Admitting this flaw and presenting the manuscript in this context would be helpful.

Validity of the findings

Results are valid and important

Additional comments

While I am in favor of publishing these results from low income countries where such data are hard to come by, the authors need to carefully proofread and edit the manuscript for concise reporting. I also think this manuscript looks at the data from different angle at what has bee already published by the same authors, but calling it retrospective is not persuading to me. I like how qPCR was used in this cohort although it is a challenging problem to distinguish if higher CT values false negative or or true negative without additional tools such as RNA-seq.

---

## Round 0.3 · accepted · Accept

Dear Authors,

Your manuscript is acceptable in its current form.

External reviews were received for this submission. These reviews were used by the Editor when they made their decision, and can be downloaded below.